# Nesting Success and Nesting Height in the Critically Endangered Medium Tree Finch (*Camarhynchus pauper*)

**Sonia Kleindorfer** [1,2,*] **, Lauren K. Common** [1] **and Petra Sumasgutner** [2]

1 College of Science and Engineering, Flinders University, Adelaide 5001, Australia; lauren.common@flinders.edu.au
2 Konrad Lorenz Research Center, Core Facility for Behavior and Cognition and Department of Behavioral and Cognitive Biology, University of Vienna, 4645 Grünau im Almtal, Austria; petra.sumasgutner@univie.ac.at
* Correspondence: sonia.kleindorfer@univie.ac.at or sonia.kleindorfer@flinders.edu.au; Tel.: +43-6604024048

**Simple Summary:** Invasive species on islands pose high risk to endemic flora and fauna unable to defend themselves. In this study, we measure the effects of introduced predators and the parasitic Avian Vampire Fly on nesting success in the critically endangered Medium Tree Finch on Floreana Island in the Galapagos archipelago. Only 10.4% of nests produced young, and the rest were killed by either predators or parasites. The lowest nests were primarily depredated by rats, the highest nests had evidence for predation by birds, and nestlings at the intermediate nest height were killed by parasitic fly larvae. In conclusion, confronted by introduced mammalian and avian predators and parasitic flies, there is no safe nesting height for this critically endangered species, and nesting success is far too low to sustain the population. Control measures are urgently needed to save Darwin's finches from extinction.

**Abstract:** When different introduced species across trophic levels (parasite, predator) invade island systems, they may pose significant threats to nesting birds. In this study, we measure nesting height and infer causes of offspring mortality in the critically endangered Medium Tree Finch (*Camarhynchus pauper*), an island endemic restricted to Floreana Island on the Galápagos Archipelago. Considering all nests at which a male built a nest, sang and attempted to attract a female (*n* = 222 nests), only 10.4% of nests produced fledglings (5% of nests had total fledging success, 5.4% of nests had partial fledging success). Of the 123 nests chosen by a female, 18.7% produced fledglings and of 337 eggs laid, 13.4% produced fledglings. Pairing success was higher for older males, but male age did not predict nesting success. All nests with chicks were infested with avian vampire fly larvae (*Philornis downsi*). We attributed the cause of death to avian vampire fly if chicks were found dead in the nest with fly larvae or pupae (45%) present. We inferred avian (either *Asio flammeus galapagoensis* or *Crotophaga ani*) predation (24%) if the nest was empty but dishevelled; and black rat (*Rattus rattus*) predation (20%) if the nest was empty but undamaged. According to these criteria, the highest nests were depredated by avian predators, the lowest nests by rats, and intermediate nests failed because of avian vampire fly larvae. In conclusion, there is no safe nesting height on Floreana Island under current conditions of threats from two trophic levels (introduced parasitic dipteran, introduced mammalian/avian predators; with Galápagos Short-Eared Owls being the only native predator in the system).

**Keywords:** *Rattus rattus*; *Philornis downsi*; Short-eared Owl; Smooth-billed Ani; avian predators; invasive species; nest condition; Galápagos Archipelago; Floreana Island; conservation management

## 1. Introduction

Islands occupy only 5.3% of the world's surface, contain a disproportionate amount of biodiversity, and experience by far the greatest proportion of species extinction and extinction risk [1], mostly due to invasive species [2,3]. Intact island systems are characterized by

high levels of endemism and few predators and parasites compared with the mainland source population [4,5], and have inspired natural history theory [6–9]. These systems are currently challenged by combinations of introduced species [10], why many endemic island species are increasingly the focus of conservation efforts [11,12]. The human-induced extinction rate is particularly high for island birds [13]. New predators and/or parasites rapidly diminish naïve prey on islands [11,14,15], either because of adult mortality or low nesting recruitment. Yet, we lack understanding into the combined effects of multiple introduced species at different trophic levels, such as parasite and predator, on avian nesting success.

Island birds tend to have higher annual survival compared with mainland counterparts [16,17]. Additionally, reproductive performance increases with age in many bird species, and females often prefer older males as partners [18,19]. Females paired with older males may receive direct benefits such as safer nest sites, increased male vigilance, and/or increased food delivery to the female [20,21]. Females paired with older males may also receive indirect benefits such as good genes for their offspring or increased male provisioning of offspring [19,22]. When introduced parasites target nestling birds but not adult birds, perhaps older parents compensate with extra food provisioning for the costs of in-nest parasitism [23]. The avian vampire fly (*Philornis downsi*) is an accidentally introduced myasis-causing fly that now occurs on the Galápagos Archipelago. The first adult vampire flies were collected on Santa Cruz Island in 1964 [24] and the first parasitic larvae were found in Darwin's Finch nests in 1997 [25]. Avian vampire fly larvae consume the blood and tissue of developing nestlings, whereas adult flies feed on decaying plant matter. A study comparing nesting success under varying conditions of avian vampire fly infestation in Small Tree Finch (*Camarhynchus parvulus*) found higher offspring survival in heavily parasitized nests when females were paired with older males rather than younger males [19]. In another study, Tree Finch parents did not increase food delivery to nests with many avian vampire fly larvae [26], a pattern that was also found in Small Ground Finch and Medium Ground Finch (*Geospiza fuliginosa* and *G. fortis,* respectively) [27,28]. Furthermore, higher nesting success in older male Small Tree Finch was explained by less predation of more concealed nests built by older males and increased predation at more exposed nests built by younger males [29]. In summary, there is evidence in several Darwin's Finch systems with nestling-only parasitism that older males have higher nesting success, but not via increased food delivery at parasitized nests.

Experience covaries with age, whereby older individuals can potentially adjust their current reproductive behaviour informed by previous breeding experience. There is some evidence that nesting height in birds is affected by prior nesting experience. For example, Pinyon Jay (*Gymnorhinus cyanocephalus*) lower their nesting height if the previous nesting attempt was depredated by a bird and build a nest in a more exposed location if the previous nesting attempt failed due to cold weather [30]. On average, experienced jays had relatively low nesting height to reduce avian predation, and they placed their nest further from the trunk early in the season to reduce incubation costs [30]. Brewer's Sparrow (*Spizella breweri*) also showed adaptive nest position in subsequent nests and in relation to prior experience [31]. Birds followed a "win:stay, lose:switch" strategy and moved further for a second nesting attempt if their previous nest had been depredated, and selected different patch attributes for the new nest [31]. In Oahu Elepaio (*Chasiempis ibidis*), an endangered Hawaiian forest bird, most low nests failed due to predation from the introduced black rat (*Rattus rattus*), but individual birds did not adjust their nesting height, although average nesting height in rat-infested areas increased 50% across a 16 year time period from 1996 to 2011 [32].

The critically endangered Medium Tree Finch (*C. pauper*) is an island endemic restricted to Floreana Island, Galápagos Archipelago. Compared with other Darwin's Finch species on Floreana, Medium Tree Finches have the most avian vampire fly larvae per nest [33], and avian vampire fly parasitism was identified as the biggest cause of nesting failure, resulting in 41% in-nest mortality during 2006 and 2008 [34]. The only nest pre-

dation studies on Floreana were carried out during 2004 and 2008 and calculated 17–44% depredated nests in Small Ground Finch [35] and 28% depredated nests in Medium Tree Finch [34]. Between 2004 and 2013, male Medium Tree Finch built nests on average 6 m above ground, which was also the trap height at which researchers caught more (egg-laying) adult female vampire flies [36]. Survey results across the decade estimated a 56.4% population decline in Medium Tree Finch between 2004 and 2008 (from 5262 to 2292 singing males), followed by a 10.7% population increase from 2008 to 2013 (from 2292 to 2537 singing males) [37]. A large-scale survey in 2015 and 2016 estimated 3900–4700 Medium Tree Finch territories [38]. Coincident with the trend for population increase in Medium Tree Finch since 2008, avian vampire fly larva, pupae and adult flies have become 6–32% smaller between 2000 and 2016, with a 26% reduction in female abdomen size [39]. However, we lack comparative studies on causes of nesting failure in this system since the last study in 2006 and 2008 [34].

The Galápagos Archipelago is influenced by the El Niño Southern Oscillation, when the Humboldt Current is interrupted and warm waters flow towards the shores of South America. El Niño lowers marine productivity, and also has strong effects on terrestrial ecosystems [40] by changing rainfall patterns and plant productivity, and thus, exerting bottom-up effects on food webs including predator–prey interactions [41]. Parasite abundance is furthermore heavily influenced by precipitation [42,43] and a recent study on Floreana Island identified rainfall, proximity to human population and fruiting trees as the most important factors determining avian vampire fly abundance [44].

The aim of this study was to identify stages and causes of nesting failure, analysing nine years of nesting data collected during 2004 to 2020 in the critically endangered Medium Tree Finch. We examined (1) effects of male age, nesting height and rainfall on stage of nesting failure (display nest, incubation, nestling period), nesting outcome (egg predation, egg abandonment, chick predation, some/all dead chicks in nest, fledge, and unknown) and causes of nesting failure (avian vampire fly larvae, avian predation, rat predation, extreme weather, unknown); and (2) if nesting height changed over time and/or correlates with male age (i.e., proxy for breeding experience). We inferred causes of nesting failure from nest contents (dead nestlings, avian vampire fly larvae) and nest condition (empty, shredded, dishevelled, missing, on ground).

## 2. Methods

### 2.1. Study Site and Study Species

The Medium Tree Finch is an island endemic restricted to Floreana Island, Galápagos Archipelago. Field work during the period February to March was conducted during 2004–2006, 2008, 2012–2014, 2016, and 2020 (Supplementary Table S1). We carried out annual surveys of nesting outcome at eight 200 m × 100 m study plots in *Scalesia*-dominated forest near Cerro Pajas (−1.299829, −90.455674) and since 2016 in 16 ha of *Scalesia* dominated forest at Asilo de la Paz (−1.313595, −90.454935). Medium Tree Finches are socially monogamous per nesting attempt. Males build a display nest and sing at the nest to attract a female. Once chosen by a female, the female brings nest lining material to complete nest construction. Clutch size is 2–4 eggs. Females are uniparental incubators and incubation lasts ~12 days. Females brood the hatchlings during the first days after hatch and both parents provision the nestlings with ~3 food deliveries per hour. The chick feeding phase lasts ~12 days [45].

### 2.2. Male Age

Male Medium Tree Finch can be aged by their plumage. Males become progressively black-hooded until attaining a fully black crown and chin from around >5 years old [29,46]. The age classifications are given as Black 0 (B0) (yearling without black plumage on crown or chin) to Black 5 (B5) (age >5 years old with fully black crown and chin) (Figure 1); females remain olive green with black streaks on their chin. Males produce a song that consists of a single syllable repeated three to seven times [37], while females are not known to

produce a song. Minimum longevity is 12 years in males (estimate based on 16 recaptures across years) with insufficient sample size to estimate minimum longevity in females (one recapture after two years) [46].

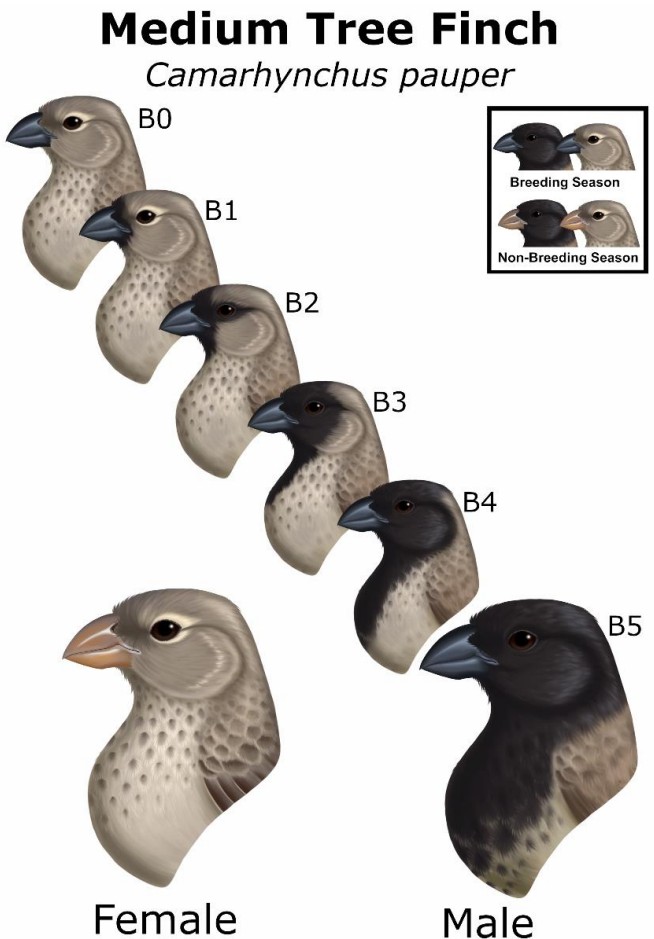

**Figure 1.** Plumage and beak characteristics in male and female Medium Tree Finch (*Camarhynchus pauper*). Beak colour in both sexes is light during the non-breeding season and black during the breeding season. Female plumage is grey-brown across their lifetime; the chin can be streaked. Yearling males are grey-brown with a streaky breast; yearling males can be distinguished by females on the basis of behaviour (e.g., song). Male crown and chin black plumage increase with annual moult from no black in yearling males (B0) to males having a fully black crown and chin from around >5 years old (B5 males). Drawing by Lauren K. Common.

*2.3. Nest Monitoring*

We monitored 238 nests and had records of the male being chosen or not for 222 of them, which features as our effective sample size. Of these nests, 92 male display nests were not chosen by a female, seven display nests were usurped by another male, and 123 nests were chosen by a female for egg-laying. Of the 123 chosen nests, nesting outcome is known for 107 nests and unknown for 16 nests. We do not analyse fledging success per egg laid in this study, but report here the total numbers: We had clutch size records from 114 out of 123 chosen nests, with a total of 337 eggs laid from which there were 217 chicks (64.4%) and 45 fledglings (13.4%).

Nests were monitored following our standardized protocol that we developed in 2000 and maintained throughout the study [33,34,47]. Nests were routinely inspected, with binoculars and ladder during 2004 to 2006, and since 2008, with a borescope, every three days during incubation and every two days during the nesting phase to confirm activity. Nesting height estimation was practiced using a laser pointer (LTI laser rangefinder) prior

to field work, which we carried out using clearly visible trees on-campus at Flinders University, Australia. The laser rangefinder was first pointed at the base of the tree and then the top to compute two vertical angles, from which tree height was calculated. We calibrated among team members at the start of the field season and visually estimated tree height as meters above ground during field work.

Nesting outcome was scored as egg predation, egg abandonment, chick predation, dead chicks in nest, fledge, and unknown. Of the 107 nests with a known outcome, 11 had 100% fledging success and 12 had partial fledging success/partial mortality. For the 96 nests with total or partial offspring mortality, we inferred the cause (avian vampire fly larvae, rat predation, avian predation, extreme weather, unknown) on the basis of nest contents (dead chicks, avian vampire fly larvae or pupae) and nest condition (empty, shredded, dishevelled, missing, on ground). We were unable to reasonably infer the cause of failure in five cases of egg abandonment. Another two cases failed due to wind, which were removed from the "cause of nest failure" analyses due to small sample size. We scored avian vampire fly as the cause of nesting failure if chicks were dead inside the nest and fly larvae or pupae were present; we scored rat predation as the cause of nesting failure if the nest contents were missing during incubation or the nestling phase (d 1 to d 8) and the nest was undamaged [48] or had a hole on top [49]; we scored avian predation as the cause of nesting failure if the nest contents were missing during incubation or the nestling phase (d 1 to d 8) and the nest was shredded, on the ground, dishevelled or missing; we scored nest failure due to extreme weather (i.e., wind) if the nesting tree had fallen down. Any other combination was scored as "unknown". We acknowledge that nest condition might be an unreliable source to infer predator identity; there are few published observations of nesting condition following nest predation and some reports are contradictory (that is, the same predator type can cause several types of post-predation nest condition) but we are consistent in our classifications, which can be tested in a future study using nest cameras [50]. From video recordings at seven Small Ground Finch (*G. fuliginosa*) nests with avian vampire flies, chick mortality in infested nests occurred in a rate of one chick per day and Darwin's Finch parents removed the dead chick if it had live siblings but left the last dead chick inside the nest [27]. Therefore, we considered avian vampire fly the cause of offspring mortality at any nest with a dead chick inside when chick age at death was 8 d or less. We used number of fledglings per nest as a measure of fledging success.

### 2.4. Mist-Netting

In our 222 analysed nests, there was a colour-banded male at 85 nest records. In 11 cases, we observed the same male across years (over two, in one case over four years); therefore, the number of unique nests is 72 with colour-banded males, including seven nests at which both the male and female were colour-banded. We mist-netted, colour-banded and measured 191 *C. pauper* between 2004 and 2020. During February of each year, for two weeks, we placed 6 × 12 m mist-nets in the study sites to capture and measure Darwin's Finches [46]. At the time of banding, each bird received a uniquely numbered aluminium band and between one to three colour-bands.

### 2.5. Statistical Analysis

All statistical analyses were performed with the software R version 4.1.2 (R Development Core Team 2021). The confidence intervals were set at 95% (corresponding to a significance level of $p = 0.05$) for all tests conducted. The Generalized Linear Models (GLMs) were fitted with the packages "lme4" [51] and "car" [52]; multinomial models were fitted with the packages "mlogit" [53] and "nnet" [54]. Model outputs were visualized with "lattice" [55], "ggplots2" [56] and "effects" [57]. No random factors were considered, as there were no considerable repeated measurements in the dataset (see section on mist-netting above). This was also reflected in an exploratory analysis at the beginning of our analytical approach, where we ran GLMMs including the random term "male ID" whereby the variable caused singularity and explained 0% of the variance, which means that the

fitted generalised linear mixed model is very close to being a generalised linear model (as indicated by comparing the glmer() model output with the glm() model output, results are very similar). In such cases, the random term should be removed [58]. Removing the random term also increased our sample size considerably. We tested for correlations of fixed effects beforehand but did not find any indication for co-linearity in our data ("year" and "annual sum of rain" only correlated with rho = −0.15). Residual distributions of the models were inspected visually to assess model fit and to validate that the model complied with the underlying assumptions (following [59]). Throughout, we report model effect sizes (estimates ± SE, derived from the respective summary functions) in the result tables and present $\chi^2$ and *p*-values based on an ANOVA Table of Deviance using Type III Wald $\chi^2$ tests (ANOVA function in "car" package) in text to report the overall significance of tested variables. We explored male age (continuous variable based on male coloration), nest height (in m) in a linear and a quadratic relationship, the interaction between male age and nest height, year, and annual sum of rain (in mm; to account for El Niño and weather effects on predation events and parasite abundance) in the full model, followed by a stepwise backward selection process [60]. We always started with the full model and simplified it using backwards elimination based on the log likelihood ratio test with $p < 0.05$ as the selection criterion (using the "drop1" command, which drops one explanatory variable, in turn, and each time applies an analysis of deviance test) until reaching the minimal adequate model. Based on the principles of parsimony (the largest amount of variance explained with the minimum number of predictors [61]), we selected the model structure that best described our response variables in the simplest manner by comparing models by deviance and the Akaike information criterion corrected for small sample sizes (AICc) after each step in the backwards elimination. Following this procedure, predictors and their interactions were excluded from final models until the lowest deviance and AICc values were reached. When candidate models had lower deviance, but higher AICc, we kept the predictor in question in the final model because this only occurred within model selections in which the difference in AICc relative to AICc min was ΔAICc < 2, and such candidate models are considered to still have substantial support [61]. In such a case, we present the final model and the backward reduced effect table in the Supplementary Material (this was only applicable for the probability of mortality analysis).

For mortality, the response variable was measured as a binary variable (0 = at least one nestling fledged, 1 = nest failed) and fitted with a binomial distribution with the logit-link function. The 11 successful nests where all chicks successfully fledged were removed from nest failure analyses; for stage of nestling mortality, the response variable was entered as an ordinal factor (0, 1, 2) and analysed using multinomial logistic regression with "0" for males not chosen or nest usurped during the courtship period (*n* = 99); "1" for nests failed during incubation (*n* = 31); and "2" for nests failed during the nestling phase (*n* = 65). For cause of nest failure, we used random utility models (within multinomial models) to analyse mortality among a set of mutually exclusive alternatives ("avian predation" *n* = 26, "vampire fly" related mortality *n* = 47 and "rat predation" *n* = 15). Two nests that failed due to strong winds and unknown causes of nest failure were excluded from this analysis. Varying sample sizes for stage of nestling mortality and causes of nestling mortality as displayed in the results section are due to incomplete records of male age and nesting height.

For nesting height (continuous response variable following a normal distribution without log transformation), we fitted a linear model with male age (continuous variable based on male coloration) and year as fixed effects, with an annual sum of rain (in mm) as an additional co-variate. The sample size for this analysis was 191 records.

## 3. Results

### 3.1. Chosen and Unchosen Males and the Stage of Nest Failure

At 222 nests at which the male built a display nest and sang to attract a female, 92 males sang but did not attract a female (41% not chosen). Older (B4 and B5) males were

2-fold more successful at attracting a female than yearling B0 or B1 males (Figure 2). Of these male display nests, seven were usurped by conspecifics. Five display nests built by yearling B0 and two display nests built by B1 males were usurped by B5 males. The multinomial model exploring the influence of male age, nesting height, year and rainfall on the stage of nest failure revealed a strong relationship with age and year, while the interaction term (male age × nesting height), nesting height (additive term) and annual rainfall did not feature in the most parsimonious model ($n$ = 188, Table 1). During the courtship period (failure due to male not chosen or nest usurped), younger males had a higher chance of failing than older males (likelihood ratio (LR) $\chi^2$ = 17.99, $p$ < 0.001; Figure 3a). The probability of failing during the courtship period also increased across study years (LR $\chi^2$ = 20.77, $p$ < 0.001; Figure 3b). No relationship was seen between failure during incubation, male age or study year. Finally, during the chick period, older males had a higher probability of failure, while, overall, offspring mortality during the chick period declined across study years.

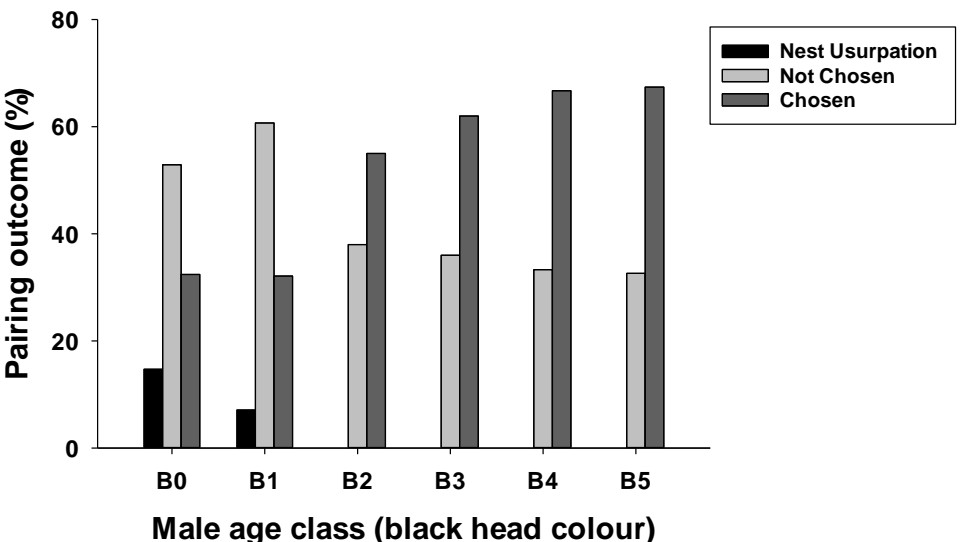

**Figure 2.** Pairing outcome is shown as the percentage of males whose nest was usurped by another male, not chosen by a female, or chosen by a female in the critically endangered Medium Tree Finch (*Camarhynchus pauper*). The percentage data are calculated within each male age class (B0 to B5; details in Figure 1). Yearling males are B0 (no black plumage) and males five years and older are B5 (fully black crown and chin). Of 222 nests at which males built a display nest and sang to attract a female, 92 males were not chosen and 7 had their nest usurped by an older conspecific. Pairing success (% males chosen by a female for nesting) was higher in older males.

**Table 1.** Multinomial logistic regression with "0" for males not chosen or nest usurped during courtship; "1" for nests failed during incubation; and "2" for nests failed during the nestling period in relation to male age (estimated by colouration) and calendar year in Medium Tree Finch (*Camarhynchus pauper*) on Floreana Island, Galápagos Archipelago. Predictors featuring significantly are displayed in bold.

| Coefficients | Estimate | SE | z-Value | *p*-Value | LR $X^2$ | *p* (>$X^2$) |
|---|---|---|---|---|---|---|
| *(Intercept): 1* | *85.07* | *101.51* | *0.84* | *0.402* | *32.65* | *<0.001* |
| *(Intercept): 2* | *360.84* | *125.84* | *2.87* | *0.004* | | |
| Male age: 1 | 0.20 | 0.13 | 1.51 | 0.132 | 17.99 | <0.001 |
| **Male age: 2** | **0.60** | **0.14** | **4.30** | **<0.001** | | |
| Year: 1 | −0.04 | 0.05 | −0.85 | 0.395 | 20.77 | <0.001 |
| **Year: 2** | **−0.18** | **0.06** | **−2.88** | **0.004** | | |

Note: Nest stage "0" = "courtship" was used as a reference category. Log-Likelihood = −135.79. Model intercept displayed in italic.

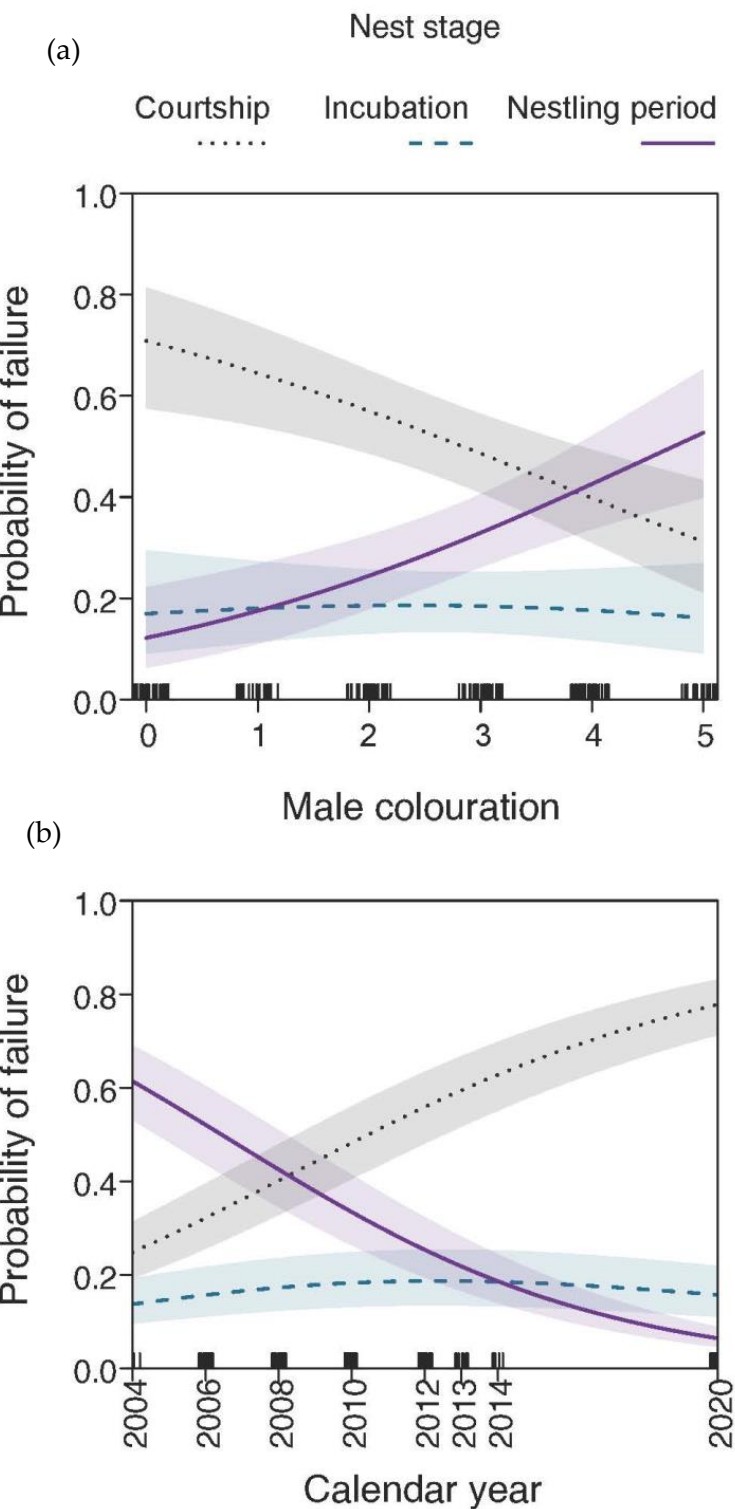

**Figure 3.** Probability of nest failure in relation to (**a**) male age (estimated by colouration as displayed in Figure 1) and (**b**) calendar year at different nest stages of Medium Tree Finch (*Camarhynchus pauper*) breeding attempts on Floreana Island, Galápagos Archipelago, between 2004 and 2020. Results show predicted line of multinomial logistic regression with 95% CIs; tick marks on x-axis reflect sample size; model details in Table 1.

### 3.2. Causes of Nesting Failure

Nesting outcome per phase (eggs, chicks) was calculated for 107 nests (16 unknown outcome) of the 123 chosen Medium Tree Finch nests (Figure 4a). Failure during incubation occurred in 31 (25%) nests due to egg predation (20%) or egg abandonment (5%). Failure during the chick period occurred in 65 (53%) nests due to chick predation (11%), all chicks dead in the nest with avian vampire fly larvae (32%), or some chicks dead in the nest with avian vampire fly larvae (10%). Total and partial fledging success occurred at 23 (19%) nests; all chicks fledged from 11 nests (9%) and some chicks fledged/died at 12 nests (10%). Nesting outcome was not known for 13% of nests.

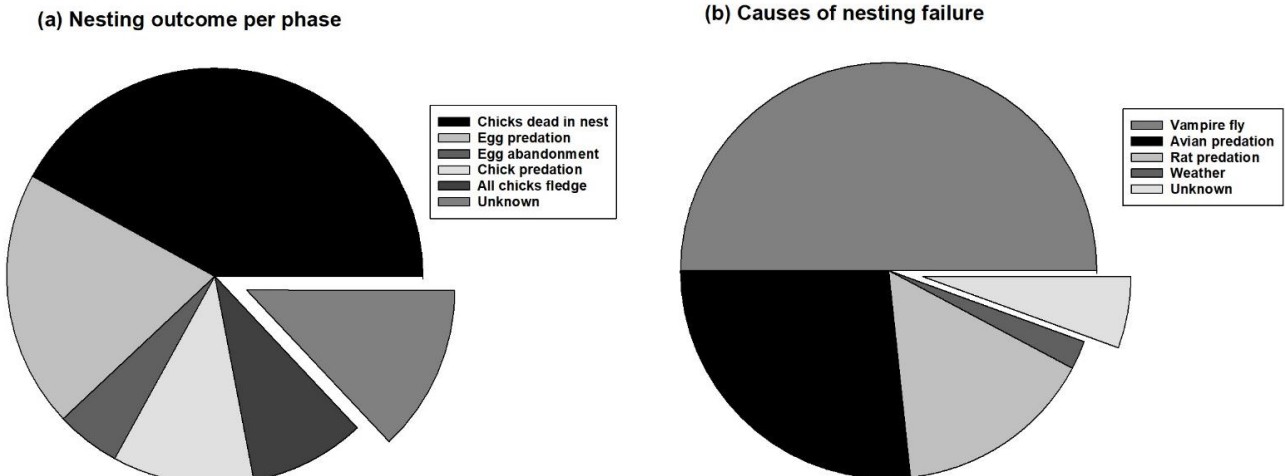

**Figure 4.** Nesting outcome (%) and causes of offspring mortality (%) in critically endangered Medium Tree Finch (*Camarhynchus pauper*) on Floreana Island monitored between 2004 and 2020. Percentage data are shown for (**a**) nesting outcome per phase (eggs, chicks) (*n* = 123), and (**b**) suspected causes of offspring mortality inferred from nest contents (eggs, chicks, avian vampire fly larvae or pupae) and nest condition (empty, dishevelled, missing) (*n* = 96). During the egg phase, offspring mortality occurred as egg predation (20%) or egg abandonment (5%). During the chick phase, offspring mortality occurred as chick predation (11%) or dead chicks in the nest (this category includes cases with all or some chicks dead in the nest) (42%). Total fledging success (all chicks fledge) occurred at 11 nests (9%) and partial fledging success/partial in-nest mortality occurred at 12 nests (10%). The cause of death inferred by nest condition was avian vampire fly parasitic larvae (45%), avian predation (24%), rat predation (14%), and extreme weather (2%). The displaced pie chart slices indicate the percentage of unknown cases (13% and 5%, respectively).

Nest condition for the 96 nests with total/partial offspring mortality was: 67 undamaged nests, 18 dishevelled or shredded nests, 1 nest with a hole on top, 3 nests on the ground, and 7 missing nests. Based on this information, and in combination with information on the age of the nest contents (eggs, chicks, avian vampire fly larvae), we estimate the cause of offspring mortality as follows (Figure 4b): in-nest dead chicks with avian vampire fly larvae attributed to avian vampire fly (45%), shredded/dishevelled/missing nest attributed to avian predation (24%), eggs or chicks missing and nest undamaged attributed to rat predation (14%), or extreme weather (2%). In 5 cases of egg abandonment, the cause of nesting failure was not known (5%).

The average ± SE number of fly larvae or pupae in nests at the time of nest collection was 50 ± 4.2 (*n* = 93 nests inspected for vampire flies). Two of 31 nests that failed during the egg stage had second instar avian vampire fly larvae (5 and 9 larvae, respectively) and all 62 nests with chicks that were inspected for vampire flies contained larvae or pupae (52 ± 4.3). The mean ± SE number of avian vampire fly larvae or pupae is given per nest and chick survival category: nests at which chicks were depredated had 23 ± 5.7 larvae/pupae, nests with total brood loss (all chicks dead in the nest) had 59 ± 6.8 larvae/pupae, nests with partial brood loss (at least one chick dead) had 55 ± 8.5 larvae/pupae, and nests from which all chicks fledged had 58 ± 9.8 larvae/pupae.

*3.3. Effects of Male Age and Nesting Height on Nesting Outcome*

The mortality analyses included only nesting attempts where the male was chosen and eggs were laid (*n* = 107 with known nestling outcome; from a total of 123 nests). The model revealed no statistically significant results (Figure S1; Table S2a) and all predictors featured into the final model (Table S2b), but note the small proportion of successful nests. Of *n* = 107 records in total, *n* = 11 fledged the full brood size (10.3%) and *n* = 12 fledged partial brood size (11.2%); therefore, *n* = 23 fledged at least one young (21.5%). When all nests are taken together, of the 222 nests at which the male built a display nest and sang to attract a female, only 10.4% successfully fledged a young, and only 5% fledged full brood sizes.

The random utility models (Table 2) exploring the three main causes of nesting failure (avian and rat predation, and avian vampire fly-related mortality) revealed a significant correlation with nesting height (quadratic term, LR $\chi^2$ = 13.37, *p* = 0.010; Figure 5a) and a non-significant trend with male age (LR $\chi^2$ = 5.65, *p* = 0.059; Figure 5b). Higher nests had a higher probability of being depredated by birds and lower nests had a higher probability of being depredated by rats. For this model (*n* = 77), the nesting height variable was missing for 10 nest records, and we removed one outlier of an exceptionally high nest (12 m); the statistical patterns remained unchanged but the removal improved model fit. Nests located at an average height (6 m) had a higher probability of being parasitized by vampire flies. Furthermore, avian vampire fly-related mortality showed a significant increase with increasing male age (partial coefficient, estimate 0.42 ± 0.19 SE, z-value = 2.20, *p* = 0.028), while no patterns between age and different forms of predation were apparent. Year and rainfall did not feature into the most parsimonious model.

**Table 2.** Random utility models (within multinomial models) among the mutually exclusive alternatives "avian predation", "avian vampire fly"-related mortality and "rat predation" as causes of nest failure in relation to male age (estimated by coloration) and nesting height (considering a linear and quadratic relationship) in Medium Tree Finch (*Camarhynchus pauper*) on Floreana Island, Galápagos Archipelago. Predictors featuring significantly are displayed in bold.

| Coefficients | Estimate | SE | z-Value | *p*-Value | LR $X^2$ | *p* (>$X^2$) |
|---|---|---|---|---|---|---|
| *(Intercept): Vampire Fly* | −0.53 | 0.66 | −0.81 | 0.420 | **19.14** | **0.004** |
| *(Intercept): Rat predation* | −0.91 | 0.84 | −1.08 | 0.279 | | |
| **Nest height (linear): Vampire Fly** | **−9.38** | **4.70** | **−1.99** | **0.046** | | |
| **Nest height (linear): Rat predation** | **−20.47** | **7.55** | **−2.71** | **0.007** | 13.37 | 0.010 |
| Nest height (quadratic): Vampire Fly | −3.34 | 4.76 | −0.70 | 0.483 | | |
| Nest height (quadratic): Rat predation | −3.94 | 6.60 | −0.60 | 0.551 | | |
| **Male age: Vampire Fly** | **0.42** | **0.19** | **2.20** | **0.028** | 5.65 | 0.059 |
| Male age: Rat predation | 0.15 | 0.25 | 0.60 | 0.548 | | |

Note: Cause of mortality "avian predation" was used as a reference category. Log-Likelihood = −68.092. Model intercept displayed in italic.

For 13 males, we had resightings across years and data on nesting height (m). Male colour (mean ± SE) was B2 (2.0 ± 0.5) for the Year 1 observation and B4 (3.6 ± 0.5) for the Year 2 observation, with on average 2.1 ± 0.6 years between the two nesting observations. There was no significant difference in nesting height across the two years (paired t-test: mean difference = −0.6 ± 0.5, 95% CI −1.18 to 0.58, t = −1.120, df = 12, *p* = 0.285). Of the 26 nesting events, we know nesting outcome in 15 cases (8 failed due to avian vampire fly parasites, 1 due to egg abandonment, 1 due to chick predation, and 5 produced fledglings). Because the resightings occurred after two years on average and nesting outcome was patchy, we did not analyse change in nesting height in relation to nesting outcome two years prior. Finally, while Medium Tree Finches build lower nests in wetter years ("annual rainfall" estimate −0.002 ± 0.0007, t = −3.00, *p* = 0.003; Figure S2), neither year nor male age featured into the minimum adequate model (Table S3).

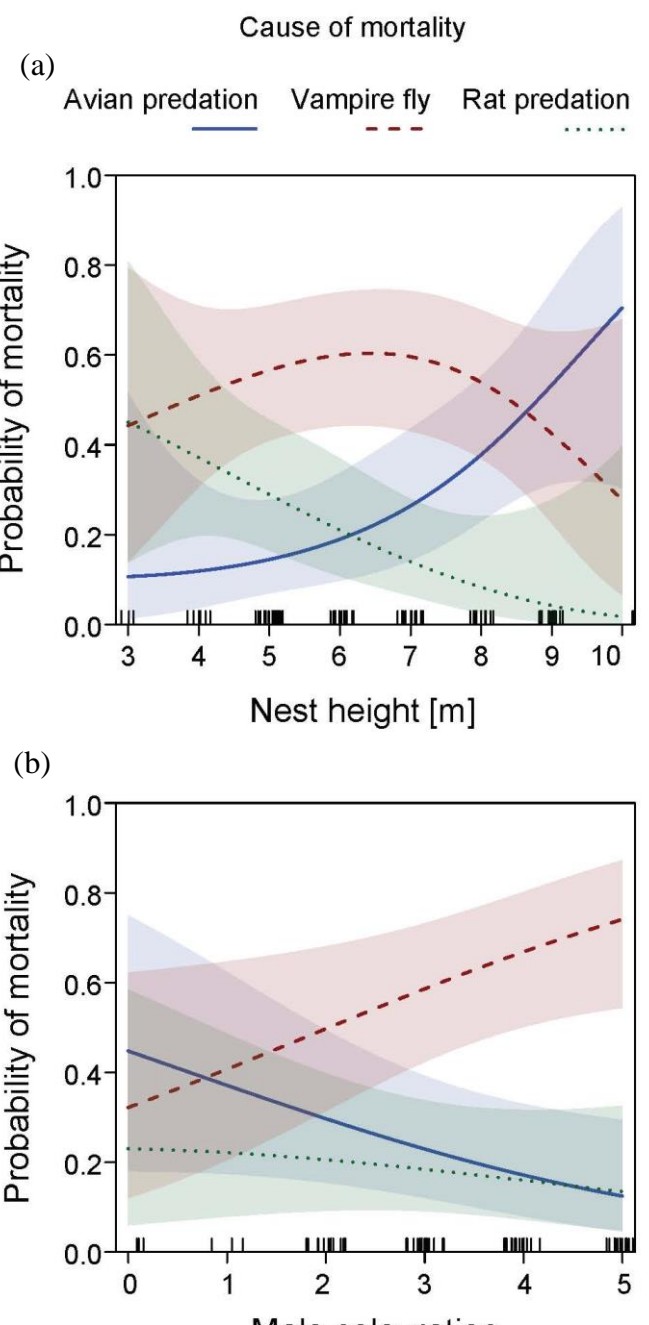

**Figure 5.** Causes of nesting failure in relation to (**a**) nesting height (quadratic term) and (**b**) male age (estimated by plumage colouration as displayed in Figure 1) in Medium Tree Finch (*Camarhynchus pauper*). Results show predicted line of random utility models (within multinomial models) among the mutually exclusive alternatives "avian predation", "avian vampire fly"-related mortality and "rat predation" with 95% CIs; tick marks on x-axis reflect sample size; model details in Table 2.

## 4. Discussion

Despite hints of population increases in the critically endangered Medium Tree Finch since 2008, the fact that 41% of males sing but do not attract a female for nesting combined with 25% egg mortality and 53% chick mortality warrant continued conservation concern and monitoring in this system. Our research aimed to identify patterns of nestling mortality across time and in relation to male age and nesting height while controlling for rainfall. Our results (1) point to higher pairing success in older males and a trend for higher nesting success across time, (2) identify avian vampire fly parasitism as the main cause

of nesting failure with evidence for high impacts from avian and rat predation, and (3) reveal different vulnerabilities to birds at different nesting heights given more suspected rat predation at lower nests, more suspected avian predation at higher nests, and more avian vampire fly parasites in nests at intermediate height. Lowest nests would be an easy find for black rats [33,62], while intermediate nests would be more susceptible to vampire flies [38], and most avian predators are visual hunters considered capable of detecting nests placed at the tips of higher branches consistent with other studies that found greater avian predation at higher nests [63,64]. Finally, (4) nesting height did not change significantly across years. While female Darwin's Finches prefer older males and their nests are more likely to survive the egg phase, they nonetheless fail during the chick phase when vampire flies target and consume chicks. In conclusion, there is no safe nesting height on Floreana Island under conditions of novel threats from two trophic levels (parasitic dipteran, avian/mammalian predators).

The Galápagos Short-eared Owl is the only remaining Galapagos endemic predator of Medium Tree Finch on Floreana Island, given that the Galápagos Hawk (*Buteo galapagoensis*) has "long been extinct" on Floreana with negligible gene flow between islands [65,66]. The Galápagos racer snake (*Pseudalsophis* spp.) no longer occurs on Floreana Island [67,68] and is furthermore not suspected of having posed a major threat to Darwin's Finches from what is known of its diet [69]. The introduced Smooth-billed Ani (*Crotophaga ani*) is an avian predator of Darwin's Finch chicks [70]. Ani were introduced to the Galápagos during the 1960s to remove ticks from introduced cattle on Santa Cruz, spread to other islands during the 1990s, and the population of ca. 250,000 now consumes Darwin's Finch chicks, endemic Galápagos cricket (large painted locust, *Gryllus abditus*), Galápagos carpenter bee (*Xylocopa darwini*), Galápagos racer snake, Galápagos lava lizard (*Microlophus albemarlensis*) and a range of other endemic fauna [70–72]. They also consume and distribute the seeds of introduced blackberry and guava, and other species, thereby contributing to significant shifts in local ecosystem function [71,73,74]. The finding that avian predation was the second leading cause of offspring mortality in Medium Tree Finch points to owl and/or ani, the only two avian predators on the island. Black rats have increased across the Galápagos Archipelago in general [75,76] and on Floreana Island in particular [77] and it is reasonable to expect that the owl population has profited [78,79] given that Short-eared Owl mostly consume rodents and birds [80]. The combination of increased rat and owl abundance could increase predation pressure on Medium Tree Finch.

The Galapagos National Park Directorate with support of Island Conservation is planning to eradicate invasive black rats and house mice (*Mus musculus*) from Floreana Island using a rodent bait formulation (Conservation 25D), a cereal pellet bait containing the rodenticide brodifacoum in 2023 [81]. Planning for non-target mortality is a cornerstone risk management component of eradication projects [82], and the mitigation plan includes a safeguard population (i.e., a small genetically representative number from different island zones) for all Darwin's Finch species and also Short-eared Owls held in captivity until traces of poison have disappeared from the environment [83,84]. Invasive rats have been eradicated from 447 islands [2,85] and, in general, the local flora and fauna have shown a rapid recovery response [81,86–92]. A planned rodent eradication from Floreana Island is expected to improve nesting success in Medium Tree Finch, though monitoring is required to measure the impact of Short-eared Owl, Smooth-billed Ani and vampire fly. Future work with camera traps would provide data on predator type within predator guilds and will be a useful tool to compare the effects of predator and parasite communities before and after mammal eradication.

Older males tend to have higher pairing success, which has been found across taxa. In crickets (Gyllididae), older males were larger and more symmetrical, and these older higher quality males had higher pairing success [93]. Males can signal their quality and/or age when male ornamentation changes with age. For example, in Yellowthroats (*Geothlypis trichas*), the colour of the black mask (melanin-based signal) increases with age but the yellow bib (carotenoid-based signal) is condition-dependent [94]; females preferred males

with brighter bibs and males with larger masks had greater social dominance. Older males that are better able to defend a good territory are predicted to have increased signalling of condition-dependent traits. Darwin's Finch males also increase the proportion of melanin-based back plumage on the chin and crown with age [29,95], yet we did not measure other traits that could covary with male age such as territory size/quality or male vocal performance. In Medium Tree Finch, younger males were less successful at attracting a female for nesting. Additionally, the proportion of younger males failing to pair increased across study years. It is possible that the operational sex ratio of Medium Tree Finch is heavily skewed in favour of males if females incur higher mortality risk from uniparental incubation. In a study on New Zealand songbirds (North Island Robin *Petroica australis longipes* and North Island Tomtit *Petroica macrocephala toitoi*), 9 of 24 territories lost breeding females mainly to black rats [48]. Given high levels of rat activity on Floreana Island, it is likely that incubating Medium Tree Finch females have higher mortality risk than nest guarding males—that is, vigilant males in the territory but not inside the nest for longer periods. It is also noteworthy that operational sex ratio biases have implications for estimating population size when surveys are conducted on singing males. In this study, older Medium Tree Finch males were more successful at attracting a female for nesting, incurred less predation from rats or avian predators, but incurred the most mortality from avian vampire flies. This perplexing pattern can be explained by the observation that the nests of older males failed the latest in the nesting cycle, when nests had the most avian vampire fly larvae that the chicks could not survive, despite the smaller size of the larvae and pupae [41].

In conclusion, this study identifies causes of nesting failure in the critically endangered Medium Tree Finch. Most younger males sing but do not attract a female for nesting (30% of B0 and B1 males pair with a female), while most older males are successful at pairing with a female (65% of B4 and B5 males pair with a female). Considering all nests at which a male built a nest, sang and attempted to attract a female (*n* = 222 nests), only 10.4% of nests produced fledglings (5% of nests had total fledging success, 5.4% of nests had partial fledging success). Considering all chosen nests with eggs (*n* = 123), 18.7% of nests produced fledglings (8.9% total success with all chicks fledged from 11 nests; 9.8% partial success with some chicks fledged and some chicks dead at 12 nests). Finally, from the perspective of all eggs laid (*n* = 337) and all fledglings produced (*n* = 45), fledging success was 13.4%. In summary, fledging success during 2004 to 2020 was between 10.4% and 18.7% in the Medium Tree Finch depending on the calculation. In combination with low average post-fledging survival of around 50% in the first month for a songbird with a chick feeding phase of 12 days [96], annual recruitment in this system is expected to be low. The additive effects of low nesting height and high nesting height as predictor variables for rat and avian predation, respectively, create an ecological trap for Medium Tree Finch who may build their nest at nesting heights that escape rat and avian predation pressure during incubation but later in the nesting cycle experience avian vampire fly parasitism. Predator and parasite control are essential to manage this critically endangered island endemic given different temporal (egg phase, chick phase) and spatial (vertical nesting height) impacts from introduced species across trophic levels.

**Supplementary Materials:** The following are available online at https://www.mdpi.com/article/10.3390/birds2040032/s1, Table S1: Sample size of monitored Medium Tree Finch (*Camarhynchus pauper*) nests on Floreana Island, Galápagos Archipelago, per year. Table S2a: Global model for probability of mortality of Medium Tree Finch (*Camarhynchus pauper*) nests on Floreana Island, Galápagos Archipelago, between 2004 and 2020, in dependency of male age (estimated by colouration) in interaction with nesting height, study year and annual sum of rain. The response variable was measured as a binary variable (0 = at least one nestling fledged, 1 = nest failed) and fitted with a binomial model with logit function; *n* = 107 records (11 complete nest successes, 12 partial successes, 84 total nest failures). Table S2b: Backward reduced effect table for probability of mortality comparing models by deviance and Akaike information criterion corrected for small sample sizes (AICc). For mortality, the null model was the minimal adequate model, but all model candidates were ranked within

(ΔAICc < 2); therefore, all predictors were kept in the final model. Figure S1: Relationship (statistically not significant) between mortality and (a) male age (estimated by colouration) in interaction with nesting height; and, (b) annual sum of rain in Medium Tree Finch (*Camarhynchus pauper*) nests on Floreana Island, Galápagos Archipelago, between 2004 and 2020. Results show predicted line of binomial model with logit function with 95% CIs, tick marks on x-axis reflect sample size, model details in Table S2a. Note that the y-axis 'nestling mortality' ranges from '0' = no mortality (all or some chicks fledged successfully) to '1' = full mortality (no chicks fledged). Table S3: Height (m) of Medium Tree Finch (*Camarhynchus pauper*) nests on Floreana Island, Galápagos Archipelago, between 2004 and 2020, in dependency of annual sum of rain; *n* = 191). Figure S2: Relationship between nesting height and annual sum of rain in Medium Tree Finches (*Camarhynchus pauper*) on Floreana Island, Galápagos Archipelago, between 2004 and 2020. Results show predicted line of linear model with 95% CIs and raw data in background scatter, model details in Table S3.

**Author Contributions:** S.K. designed the research and wrote the first draft of the paper; S.K. and L.K.C. collected the data; S.K. and P.S. analysed the data; all authors edited the manuscript. All authors have read and agreed to the published version of the manuscript.

**Funding:** The study was financially supported by the Australian Research Council (LP0991147, DP190102894), Rufford Small Grant Foundation, Mohamed bin Zayed Species Conservation Fund, Max Planck Institute for Ornithology, Royal Society for the Protection of Birds/Birdfair, and Galápagos Conservation Trust.

**Institutional Review Board Statement:** This research is approved by Flinders University Animal Welfare Committee E393 and E480 and Ministry of the Environment, Ecuador MAE-DNB-CM-2016-0043. This descriptive study is classified as non-animal experiments in accordance with the Austrian Animal Experiments Act (§ 2. Federal Law Gazette No. 501/1989).

**Data Availability Statement:** The data for this manuscript are available on Dryad https://doi.org/10.5061/dryad.jm63xsjcd.

**Acknowledgments:** Permission to conduct this study was granted by the Galápagos National Park Directorate (DPNG) (PC-021-99, PC-19-07, PC-39-09, PC-58-11, PC-38-12, PC-15-14, PC-23-16, PC-02-20) with logistical support provided by the Charles Darwin Research Station (CDRS). We thank Christian Sevilla, Edison Muñoz, and Edgar Masaquiza for support from the DPNG on Santa Cruz and Edie Rocero, Luis Alexander Araujo, Anibal Altamirano, Anibal San Miguel and Wilma Pérez from the DPNG on Floreana Island. Logistical support for the project was provided by the CDRS and we thank Marta Romoleroux, Nicolas Padilla and Mercy Torres. We thank the community of Floreana for their support, with special thanks to Walter Cruz, Claudio Cruz, and Maria de Lourdes Soria, Ingeborg and Erika Wittmer. We thank members of the *Philornis downsi* Action Plan and Galápagos Land Bird Project for constructive planning, in particular Charlotte Causton and Birgit Fessl. For field assistance, we thank J. O'Connor, R. Dudaniec, R. Christensen, J. Garcia Loor, C. Akçay, D. Colombelli-Négrel, A. Katsis, A. Hohl, L. Hohl, D. Arango Roldan, V. Puehringer-Sturmeyr, M. Gallego-Abenza, K. Peters, C. Evans, and M. Louter. We wish to thank N. Adreani for discussions on the statistical approach. This publication is contribution number 2429 of the Charles Darwin Foundation for the Galápagos Islands.

**Conflicts of Interest:** The authors declare no competing interests.

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
