# Peer review of "Nesting Success and Nesting Height in the Critically Endangered Medium Tree Finch (Camarhynchus pauper)"

_2673-6004, doi:10.3390/birds2040032_

Round 1
Reviewer 1 Report
Review comments
In this manuscript, Kleindorfer and colleagues report the results from a correlational study on causes of nesting failure and offspring mortality in medium tree finch on Floreana Island (Galápagos), with focus on nesting height and male age. They collected nesting data in nine years during 2004-2020, in two sub-areas in relatively close proximity. The authors found that both courtship and nesting success was generally low, and that all nesting heights suffered high offspring mortality, with avian predators depredating the highest nests, rats depredating the lowest, and avian vampire fly larvae causing mortality in the intermediate heights. The tree-height estimation method (visual estimation following prior practice) clearly introduces possible biases. That being said, when used as a rough measure to estimate height as done here, it should still be deemed a functional method.
The study species is critically endangered and the work is contributing to knowledge that may both aid conservation actions and increase focus on the matter. The authors generally present a well-written, interesting and structured manuscript (with some exceptions, see below). However, there are certain major issues that need to be addressed, in particular in the methodology. Perhaps most important are the concerns about the model selection procedure, which, if unsolved, would give less confidence in the results presented.
Major points:
Summary:
The sentence in the summary (line 13) “Only 10.4% of nests produced young, and the rest were killed by either predators or parasites” gives the impression that 89.6% of nests were predated, while the de facto predation rate on nests with eggs was much lower (78%). This should be rephrased/clarified.
Introduction:
The authors demonstrate solid knowledge of the literature, where the rationale for the research questions involving male age and nest height are well founded in the introduction. The background on vertebrate nest predators (in particular avian predators) seems somewhat less developed, and a bit unbalanced in comparison with the thorough background (and literature references) on vampire fly. Further, the section presenting the research questions in the end of the introduction leaves the reader puzzled about why rainfall is introduced as a predictor of nesting failure, with no mentioned of this beforehand. Although previous studies have related rainfall to nesting success, this should be presented earlier as a rationale for the inclusion of this variable.
The aims of the study are partially presented in line 49-51, and then again in the end of the Introduction. The purpose of the study would be clearer and easier to follow if aims were presented only in one place.
On line 107-108, the methodology chosen for deciding causes of nesting failure seems more correct to be part of the Methods section – and also, it should be elaborated (before the Discussion) why the different nest conditions are decided to point to a certain conclusion.
Methods:
L123-137, including Fig. 1: The male ages are shown on a scale from B0 to B5, where B0 is defined as “yearlings” and B5 is defined as “5+ years old”. If Fig. 1 shows adult plumage in intervals of one year, then B5 would be adults of six years and up. The term “5+” is easily interpreted as “5 or more”, thus, I would suggest writing “>5 years old” if this is the case. Note also that the x-axis in Fig. 2 shows male age from “0” to “5” (presumably referring to the groups “B0” to “B5” and not to age?), adding to the confusion on the matter.
L140: How many nests were monitored? Here you state 238, but adding up the numbers in the next sentence points to 222 being the correct number of nests. See also L178, stating 238 nests. Have any nests been filtered out before analyses? (And if so, what is the reason for this?).
You monitored a certain amount of nests, but from Fig. 3 it appears that there was much variance in sample sizes among years. I would appreciate a table stating sample sizes in each year. Is the sample size in all years high enough to make representative comparisons between years?
L190: You do not consider random effects, but as there were 14 cases of repeated monitoring of the same males, there is clearly a chance that there is independence that should have been accounted for. It might be ok to analyse without random effects, but I would recommend testing for this (or at least inspecting the grouped data) first to ensure that you are not treating dependent data as independent.
L198-202: There is some important information on model selection that is missing in the description of methods and, consequently, in the results section. When using AIC/AICc, there is a clear recommendation from the authors already cited in the manuscript (Burnham & Anderson, several articles and books) that models with AIC-values in close proximity to each other should be seen as equally good. Thus, model selection by AIC should be presented with information (e.g. a table) showing the AIC scores of all models (or at least all top models and the null model), with a defined (selected) threshold for when models are deemed equal. Without this information it is not possible to judge whether the top model is really a single top model, or one of several models that describe the relationships equally well.
L292-297: I do not agree with the authors that several of the results of the mortality analyses can be said to show “slight indications”, when the uncertainty estimates are so large that confidence intervals are crossing zero by a large margin. Earlier in the manuscript a significance level of p=0.05 is mentioned, and all these estimates are far from this level. Also, for some reason, none of the p-values referred to in the main text here are consistent with the p-values shown in the supplementary material for this analysis. I am also concerned about the statement that all predictors were present in the most parsimonious model. With several predictors clearly without relevance, one would expect that this model had higher AIC score (thus, being a poorer fit to the data) than competing models without these predictors (avoiding the penalties of the excess variables). I would recommend assuring that there are no missing values in any of the predictors before comparing models with AIC, as AIC model selection depends completely on all values being present in all models that are compared. Also, model selection tables including at least AIC values and number of parameters in the different models should be presented (see above), either in the manuscript or in the supplementary material.
L332-333: Also here I think some more information about the method would improve clarity. In Table S2 you present a single model with all predictors, and as it is not mentioned I assume there was no model selection in this analysis (if so, why not, as this is performed elsewhere?). Presenting a model including clearly unrelated parameters affects the final estimates for the one parameter that seems to have an impact on the response. I recommend to perform a model selection with presentation of model selection tables and estimates for the best model(s), or, explain why model selection is not performed in this analysis. Also, on the detail side, note that a) the estimate and SE for Rain in Table S2 are uninformative, showing no biological effects unless more decimals are added (which is warranted in this case since the variable scale is in hundreds of mm), further, b) the t-value in the manuscript text is rounded in the wrong direction, and c) there is no notation (a/b/c) in Fig. S2 although this notation is used in the figure caption.
Discussion:
The study have many interesting findings, but I miss a thorough discussion of one of them: It seems that the probability of failed courtship has increased drastically over the years, but that the probability of a failed nestling period has decreased equally drastically. What could explain this pattern? (Could this relate to changes in the ecosystem or management actions over the years?). These seem to be important questions and I believe the manuscript would benefit from addressing them clearer.
Minor comments:
L31: “Rattus rattus” should be in italics
L45: Misplaced comma after “endemic island species”
L57: Older parents compensate how? This is currently unexplained in the Introduction
L66: Misplaced word, “respectively” refers to what is inside the parenthesis and should be placed within the brackets
L100: Lack of studies on nesting failure, not lack of data (which is collected for the current study)?
L104: Although “display nest” is a pre-mating signal known from a number of bird species, the term should be briefly presented for the readers
L114: ‘carry’ should be ‘carried’
L116: “Finch” should here be “Finches”
L137: “Minimum longevity in males is 12 years (Langton and Kleindorfer 2019).” This is a repeated sentence from the main text, and not directly related to the figure. Also, the reference style here is inconsistent.
L147: “are” should be “were”
L156-161: I strongly recommend splitting up and rephrasing this sentence for clarity...
L171, L179, L180, L223: Numbers under 10 should normally be spelled out
L197: Why annual sum of rain instead of breeding-time sum of rain? This deserves an explanation.
L226: There is no notation of a) and b) in the figure.
L292: Note that Fig. S1b is not presented in the supplementary material, but it is still referred to in the figure caption…
L322: There seems to be a mix-up with quadratic/linear nest height in Table 2.
L332: I would recommend consistency in species names when it comes to capital first letters. Here “medium tree finches” is not capitalized, other places it is. Other species names are also written both with and without capital first letter (e.g. L74 and L77).
L359, L360 and other places in the discussion: Scientific names are missing at first mention of species.
L385: “younger males were less successful at attracting a female for nesting, and their pairing failure increased across study years.” This statement sounds like the pairing failure of the younger males increased over years, but to my understanding the authors only looked at whether pairing failure increased in general over years, not in interaction with male age.
Author Response
Responses to Reviewer 1
In this manuscript, Kleindorfer and colleagues report the results from a correlational study on causes of nesting failure and offspring mortality in medium tree finch on Floreana Island (Galápagos), with focus on nesting height and male age. They collected nesting data in nine years during 2004-2020, in two sub-areas in relatively close proximity. The authors found that both courtship and nesting success was generally low, and that all nesting heights suffered high offspring mortality, with avian predators depredating the highest nests, rats depredating the lowest, and avian vampire fly larvae causing mortality in the intermediate heights. The tree-height estimation method (visual estimation following prior practice) clearly introduces possible biases. That being said, when used as a rough measure to estimate height as done here, it should still be deemed a functional method.
The study species is critically endangered and the work is contributing to knowledge that may both aid conservation actions and increase focus on the matter. The authors generally present a well-written, interesting and structured manuscript (with some exceptions, see below). However, there are certain major issues that need to be addressed, in particular in the methodology. Perhaps most important are the concerns about the model selection procedure, which, if unsolved, would give less confidence in the results presented.
# Author’s response: Thank you very much for your helpful assessment. We clarified the analytical approach and addressed all other issues that were pointed out as detailed below.
The sentence in the summary (line 13) “Only 10.4% of nests produced young, and the rest were killed by either predators or parasites” gives the impression that 89.6% of nests were predated, while the de facto predation rate on nests with eggs was much lower (78%). This should be rephrased/clarified.
# Author’s response: As stated in the Discussion, of 222 nests only 10.4% produced fledglings. If one excludes those nests not chosen by females, then, of the 123 chosen nest with eggs, 18.7% produced fledglings and 25% failed during the egg phase and 53% failed during the chick stage (2% unknown). From the perspective of all eggs laid (N = 337) and all fledglings produced (N = 45), fledging success was 13.4%. We have rewritten the abstract to reduce confusion. We removed the stage of nesting failure form the abstract and instead report the three calculations of fledgling success. New text reads: “Considering all nests at which a male built a nest, sang and attempted to attract a female (N = 222 nests), only 10.4% of nests produced fledglings (5% of nests had total fledging success, 5.4% of nests had partial fledging success). Of the 123 nests chosen by a female, 18.7% produced fledglings and of 337 eggs laid, 13.4% produced fledglings.”
The authors demonstrate solid knowledge of the literature, where the rationale for the research questions involving male age and nest height are well founded in the introduction. The background on vertebrate nest predators (in particular avian predators) seems somewhat less developed, and a bit unbalanced in comparison with the thorough background (and literature references) on vampire fly. Further, the section presenting the research questions in the end of the introduction leaves the reader puzzled about why rainfall is introduced as a predictor of nesting failure, with no mentioned of this beforehand. Although previous studies have related rainfall to nesting success, this should be presented earlier as a rationale for the inclusion of this variable.
# Author’s response: The reviewer has correctly identified much research has focused on the avian vampire fly that was accidentally introduced to the Galápagos Islands, but to date hardly any nest predation studies have been done. This is exactly the research gap we are aiming to fill – but there is simply not much literature on the topic to cite at the moment. We added some more information on other systems, specifically regarding relationships between precipitation and predation (bottom-up effects like for example demonstrated in relation to El Niño).
The aims of the study are partially presented in line 49-51, and then again in the end of the Introduction. The purpose of the study would be clearer and easier to follow if aims were presented only in one place.
# Author’s response: We removed the highlighted section and incorporated the cited literature at relevant places elsewhere, to have one dedicated aims and predictions section at the end of the introduction.
On line 107-108, the methodology chosen for deciding causes of nesting failure seems more correct to be part of the Methods section – and also, it should be elaborated (before the Discussion) why the different nest conditions are decided to point to a certain conclusion.
# Author’s response: We left this sentence in the introduction because this is a key element of the study: we infer from nest condition the possible causes of nest failure, and we would like to be up front with the reader.
L123-137, including Fig. 1: The male ages are shown on a scale from B0 to B5, where B0 is defined as “yearlings” and B5 is defined as “5+ years old”. If Fig. 1 shows adult plumage in intervals of one year, then B5 would be adults of six years and up. The term “5+” is easily interpreted as “5 or more”, thus, I would suggest writing “>5 years old” if this is the case. Note also that the x-axis in Fig. 2 shows male age from “0” to “5” (presumably referring to the groups “B0” to “B5” and not to age?), adding to the confusion on the matter.
# Author’s response: We corrected this throughout.
L140: How many nests were monitored? Here you state 238, but adding up the numbers in the next sentence points to 222 being the correct number of nests. See also L178, stating 238 nests. Have any nests been filtered out before analyses? (And if so, what is the reason for this?).
# Author’s response: We added a line in the nest monitoring paragraph to clarify: “We monitored 238 nests, and had records of the male being chosen or not for 222 of them, which features as our effective sample size.” We also changed the paragraph mist-netting and only refer to the 222 nests that were relevant for the analysis here (thus, detailed numbers on colour ringed males and repeated measures have changed slightly as we used 222 and not 238 as the reference).
You monitored a certain amount of nests, but from Fig. 3 it appears that there was much variance in sample sizes among years. I would appreciate a table stating sample sizes in each year. Is the sample size in all years high enough to make representative comparisons between years?
# Author’s response: We added another supplementary table clarifying the effective sample sizes per year (see revised Table S1). We also noted when inspecting Figure 3 that year 2005 was displayed there without any data collection taking place. We removed the year from the figure. It correctly displays data for 2004, 2006, 2008, 2010, 2012-2014 and 2020 now. The smallest sample size is 2004 with 5 nests only, followed by 2014 with 9 nests. These data points do not skew the general temporal pattern we see between 2004 and 2020.
L190: You do not consider random effects, but as there were 14 cases of repeated monitoring of the same males, there is clearly a chance that there is independence that should have been accounted for. It might be ok to analyse without random effects, but I would recommend testing for this (or at least inspecting the grouped data) first to ensure that you are not treating dependent data as independent.
# Author’s response: We did inspect group data before going into the analysis and did not see any confounding effects. We only have a small fraction of nesting records with colour ringed males (n=85) so the sample size for the analysis including a random factor would be very small and is not comparable with the larger data set (n=222). With 11 repeated measures out of 85 nest records it is still a small proportion (of 12%, with two breeding attempts per male, which could induce some pseudoreplication). However, variance of this random effect (when considered) was 0. This supports that, although biologically breeding attempts of the same breeder are not independent, statistically (at least in our data set), they are. We clarified this in the methods now (statistical analysis section) which now reads: “No random factors were considered, as there were no considerable repeated measurements in the data set (see section mist netting above). This was also reflected in an exploratory analysis at the beginning of our analytical approach, where we run GLMMs including the random term ‘male ID’ whereby the variable caused singularity and explained 0% of the variance, which means that the fitted generalised linear mixed model is very close to being a generalised linear model (as indicated by comparing the glmer() model output with the glm() model output, results are very similar). In such cases the random term should be removed (Pasch et al. 2013 Am Nat). Removing the random term also increased our sample size considerable.”
L198-202: There is some important information on model selection that is missing in the description of methods and, consequently, in the results section. When using AIC/AICc, there is a clear recommendation from the authors already cited in the manuscript (Burnham & Anderson, several articles and books) that models with AIC-values in close proximity to each other should be seen as equally good. Thus, model selection by AIC should be presented with information (e.g. a table) showing the AIC scores of all models (or at least all top models and the null model), with a defined (selected) threshold for when models are deemed equal. Without this information it is not possible to judge whether the top model is really a single top model, or one of several models that describe the relationships equally well.
# Author’s response: The reviewer must have misunderstood. We did not test a set of candidate models (as in an information theoretic approach). Instead, a stepwise backward selection process was applied following the recommendations by Zuur et al 2009 (we cited) in the model selection section of the book (page 220-222). We clarified this and extended on the Statistical analysis section to avoid any further confusion: “We always started with the full model and simplified it using backwards elimination based on log likelihood ratio test with P<0.05 as the selection criterion (using the ‘drop1’ command, which drops one explanatory variable, in turn, and each time applies an analysis of deviance test) until reaching the minimal adequate model.”
L292-297: I do not agree with the authors that several of the results of the mortality analyses can be said to show “slight indications”, when the uncertainty estimates are so large that confidence intervals are crossing zero by a large margin. Earlier in the manuscript a significance level of p=0.05 is mentioned, and all these estimates are far from this level. Also, for some reason, none of the p-values referred to in the main text here are consistent with the p-values shown in the supplementary material for this analysis. I am also concerned about the statement that all predictors were present in the most parsimonious model. With several predictors clearly without relevance, one would expect that this model had higher AIC score (thus, being a poorer fit to the data) than competing models without these predictors (avoiding the penalties of the excess variables). I would recommend assuring that there are no missing values in any of the predictors before comparing models with AIC, as AIC model selection depends completely on all values being present in all models that are compared. Also, model selection tables including at least AIC values and number of parameters in the different models should be presented (see above), either in the manuscript or in the supplementary material.
# Author’s response: We revised the paragraph as suggested and simply state that there was no relationship apparent with any of the tested predictor variables.
L332-333: Also here I think some more information about the method would improve clarity. In Table S2 you present a single model with all predictors, and as it is not mentioned I assume there was no model selection in this analysis (if so, why not, as this is performed elsewhere?). Presenting a model including clearly unrelated parameters affects the final estimates for the one parameter that seems to have an impact on the response. I recommend to perform a model selection with presentation of model selection tables and estimates for the best model(s), or, explain why model selection is not performed in this analysis. Also, on the detail side, note that a) the estimate and SE for Rain in Table S2 are uninformative, showing no biological effects unless more decimals are added (which is warranted in this case since the variable scale is in hundreds of mm), further, b) the t-value in the manuscript text is rounded in the wrong direction, and c) there is no notation (a/b/c) in Fig. S2 although this notation is used in the figure caption.
# Author’s response: Table S2 still displays the most parsimonious model after stepwise backward elimination as described in the method, and includes male age, study year and rain – dropping any of these predictors in the applied stepwise backward elimination procedure did not improve deviance or AICc. They all explain some of the variance of the data, but only rain features significantly.
The study have many interesting findings, but I miss a thorough discussion of one of them: It seems that the probability of failed courtship has increased drastically over the years, but that the probability of a failed nestling period has decreased equally drastically. What could explain this pattern? (Could this relate to changes in the ecosystem or management actions over the years?). These seem to be important questions and I believe the manuscript would benefit from addressing them clearer.
# Author’s response: We did not engage in this here as it would introduce too much speculation. Rather, we will continue to measure and monitor the change in nesting and pairing patterns over the next years, also during the rat and feral cat eradication, and we will address this in a focused manuscript in the future.
Minor comments:
# Author’s response: Thank you for these details, we have corrected them throughout.
L31: “Rattus rattus” should be in italics
L45: Misplaced comma after “endemic island species”
L57: Older parents compensate how? This is currently unexplained in the Introduction
L66: Misplaced word, “respectively” refers to what is inside the parenthesis and should be placed within the brackets
L100: Lack of studies on nesting failure, not lack of data (which is collected for the current study)?
L104: Although “display nest” is a pre-mating signal known from a number of bird species, the term should be briefly presented for the readers
L114: ‘carry’ should be ‘carried’
L116: “Finch” should here be “Finches”
L137: “Minimum longevity in males is 12 years (Langton and Kleindorfer 2019).” This is a repeated sentence from the main text, and not directly related to the figure. Also, the reference style here is inconsistent.
L147: “are” should be “were”
L156-161: I strongly recommend splitting up and rephrasing this sentence for clarity...
L171, L179, L180, L223: Numbers under 10 should normally be spelled out
L197: Why annual sum of rain instead of breeding-time sum of rain? This deserves an explanation.
L226: There is no notation of a) and b) in the figure.
L292: Note that Fig. S1b is not presented in the supplementary material, but it is still referred to in the figure caption…
L322: There seems to be a mix-up with quadratic/linear nest height in Table 2.
L332: I would recommend consistency in species names when it comes to capital first letters. Here “medium tree finches” is not capitalized, other places it is. Other species names are also written both with and without capital first letter (e.g. L74 and L77).
L359, L360 and other places in the discussion: Scientific names are missing at first mention of species.
L385: “younger males were less successful at attracting a female for nesting, and their pairing failure increased across study years.” This statement sounds like the pairing failure of the younger males increased over years, but to my understanding the authors only looked at whether pairing failure increased in general over years, not in interaction with male age.

Reviewer 2 Report
Dear Authors,
Many thanks for submitting an engaging research paper on the causes of mortality of medium tree finches. I found the work to be interesting and there is a clear conservation value to the work. In light of this, I suggest that the paper has some merit as a publication.
There are some revisions required and I have added specific comments on the pdf version of the manuscript. Additionally, please consider the following key areas:
- Assumptions. The cause of mortality for many of the chicks has been assumed based on environmental signs. While it is difficult to determine the exact cause of death in many cases, the assumption that, for example, rats were the culprit, does bring some uncertainty into the findings. This potential uncertainty needs to be acknowledged more clearly. For example, when inferring that a certain death took place based only on an empty nest, I would like to see some consideration that other causes other than rats may have resulted in the disappearance of the chicks.
- Species names. A more minor point, but check through for consistency of use of species names. Generally, the common and scientific name should be used at the first mention of the species, and from then on only the common or scientific name is needed. In some areas of the manuscript, a blend of techniques are used.
- Proof reading. There remain some small grammar errors in the work that could be easily fixed,
With these revisions, I am confident that the manuscript will be in a good position overall.

Author Response
Responses to Reviewer 2
Many thanks for submitting an engaging research paper on the causes of mortality of medium tree finches. I found the work to be interesting and there is a clear conservation value to the work. In light of this, I suggest that the paper has some merit as a publication.
# Author’s response: Thank you very much for the kind assessment of our work.
There are some revisions required and I have added specific comments on the pdf version of the manuscript. Additionally, please consider the following key areas:
Assumptions. The cause of mortality for many of the chicks has been assumed based on environmental signs. While it is difficult to determine the exact cause of death in many cases, the assumption that, for example, rats were the culprit, does bring some uncertainty into the findings. This potential uncertainty needs to be acknowledged more clearly. For example, when inferring that a certain death took place based only on an empty nest, I would like to see some consideration that other causes other than rats may have resulted in the disappearance of the chicks.
Species names. A more minor point, but check through for consistency of use of species names. Generally, the common and scientific name should be used at the first mention of the species, and from then on only the common or scientific name is needed. In some areas of the manuscript, a blend of techniques are used.
Proof reading. There remain some small grammar errors in the work that could be easily fixed,
With these revisions, I am confident that the manuscript will be in a good position overall.
# Author’s response: We checked all species names throughout and corrected the smaller errors.
Additionally, we addressed all minor issues that were outlined in the enclosed PDF and would like to thank the reviewer for the effort.

Reviewer 3 Report
The issue is cogent and may raise some interest in the readership of this avian journal. The conservation background and aims make it worth of publishing.
Non all species and Genera names are reported in italics. This is a very severe relevant, Linneian mistake, therefore they do need to be precisely corrected throught the entire text.
Specific ponts:
367 The Galapagos National Park Directorate with support of Island Conservation is planning to eradicate invasive black 368 rats and house mice from Floreana Island using a rodent bait formulation (Conservation 25D), a cereal pellet bait con- 369 taining the rodenticide brodifacoum in 2023 [76]. Invasive rats have been eradicated from 447 islands [2,77] and in
1) the poisonous baits are risky for nontarget species, should me clearly stated. More in general, some "operative" reflections for environmental management emerged from the present paper data should be concisely added as conclusion, possibly in a separate, final paragraph.
have higher mortality risk than nest guarding males. It is also noteworthy that operational sex ratio biases have impli- 391
2) Some definition of "nest guarding males" seems necessary.
had greater social dominance. Older males that are better able to defend a good territory are predicted to have in- 382
2) A clear definition of "greater social dominance" seems indeed necessary. Should also include a brief description of the social oranization od this bird species. Available literature needs to be enlisted.
Author Response
Responses to Reviewer 3
The issue is cogent and may raise some interest in the readership of this avian journal. The conservation background and aims make it worth of publishing.
Non all species and Genera names are reported in italics. This is a very severe relevant, Linneian mistake, therefore they do need to be precisely corrected through the entire text.
# Author’s response: We checked all species names throughout and corrected them regarding capitals and italics as required.
Specific points:
367 The Galapagos National Park Directorate with support of Island Conservation is planning to eradicate invasive black 368 rats and house mice from Floreana Island using a rodent bait formulation (Conservation 25D), a cereal pellet bait con- 369 training the rodenticide brodifacoum in 2023 [76]. Invasive rats have been eradicated from 447 islands [2,77] and in
1) the poisonous baits are risky for nontarget species, should me clearly stated. More in general, some "operative" reflections for environmental management emerged from the present paper data should be concisely added as conclusion, possibly in a separate, final paragraph.
# Author’s response: We have added a section on this in the discussion: “Planning for non-target mortality is a cornerstone risk management component of eradication projects [Travers et al (2021): Bottom‐up effect of eradications. J. Appl. Ecol. Doi:10.1111/1365-2664.13828.], and the mitigation plan includes a safeguard population (i.e., a small genetically representative number from different island zones) for all Darwin’s Finch species and also Short-eared Owls held in captivity until traces of poison have disappeared from the environment [Island-Conservation. Floreana island ecological restoration: Rodent and cat eradication feasibility analysis (Santa Cruz, California, 2013); Rodríguez & Fessl. Population viability analysis to assess the impact of non-target mortality in Floreana Island, Galapagos. Conservation Breeding Specialist Group (Santa Cruz, Galapagos, 2016)]”
have higher mortality risk than nest guarding males. It is also noteworthy that operational sex ratio biases have impli- 391
2) Some definition of "nest guarding males" seems necessary.
# Author’s response: we added the following text: “– that is, vigilant males in the territory but not inside the nest for longer periods”.
had greater social dominance. Older males that are better able to defend a good territory are predicted to have in- 382
3) A clear definition of "greater social dominance" seems indeed necessary. Should also include a brief description of the social organization od this bird species. Available literature needs to be enlisted.
# Author’s response: this information can be found in the cited study ref #86

Round 2
Reviewer 1 Report
The revised manuscript is generally much improved from the first version. The comments in the first review have mostly been responded to either by suitable changes in the manuscript or good explanations in the authors’ responses. My main concern about the manuscript, regarding the model selection procedure, was also addressed, although I still believe that this section could be further improved. The authors claim in their response that there must be a misunderstanding, but the first version clearly stated that model selection was performed by ranking candidate models. With the updated methods description it is stated that “drop1” was the approach used for model selection, but as the method includes AIC I still miss information showing how the different models performed in competition with each other (see previous review for details). As the authors e.g. present “most parsimonious models” with all predictors included, even when these are highly insignificant and less likely to be good descriptors of the relationships, more transparency would give more confidence in the results and applied methods.
Additionally, as pointed out in the first review, the x-axis in Fig. 2 is misleading. It currently claims to show “male age (black head colour)”, but what it really shows is the constructed groups B0-B5. The text description of ages has been corrected since the first review, but not Fig. 2. Furthermore, the y-axis in Fig. 2 is now changed since the first version. Where it previously only said “percentage”, it now says “% paired males” – but as I understand it, what the figure aims to show is the percentage of paired vs unpaired males over the years. Thus, the y-axis does not show the percentage of paired males, but rather the percentage of males (where some were paired and some unpaired).
Author Response
The revised manuscript is generally much improved from the first version. The comments in the first review have mostly been responded to either by suitable changes in the manuscript or good explanations in the authors’ responses. My main concern about the manuscript, regarding the model selection procedure, was also addressed, although I still believe that this section could be further improved. The authors claim in their response that there must be a misunderstanding, but the first version clearly stated that model selection was performed by ranking candidate models. With the updated methods description it is stated that “drop1” was the approach used for model selection, but as the method includes AIC I still miss information showing how the different models performed in competition with each other (see previous review for details). As the authors e.g. present “most parsimonious models” with all predictors included, even when these are highly insignificant and less likely to be good descriptors of the relationships, more transparency would give more confidence in the results and applied methods.
OUR RESPONSE: We did not realise that the reviewer was referring to model results in the supplementary material. We have added the required information now. When using ‘drop1’, both, model deviance and AIC are computed and can be used to determine the minimum adequate model. This process becomes challenging in top ranked models within ΔAICc <2, because deviance might be lower, but AICc might be slightly (within ΔAICc <2) higher, which still indicates high support for the model as pointed out by the reviewer before. In such a case (which applied to our ‘probability of mortality’ model that suffered from small sample size and did not reveal any significant results – thus, the table and figure are only displayed in supplementary material) we kept the predictor in question in the final model. We added this information in the revised methods (lines 310-315) and added the backward reduced effect table in the supplementary material. We also changed the wording from most parsimonious model to minimum adequate model to be more accurate.
Additionally, as pointed out in the first review, the x-axis in Fig. 2 is misleading. It currently claims to show “male age (black head colour)”, but what it really shows is the constructed groups B0-B5. The text description of ages has been corrected since the first review, but not Fig. 2. Furthermore, the y-axis in Fig. 2 is now changed since the first version. Where it previously only said “percentage”, it now says “% paired males” – but as I understand it, what the figure aims to show is the percentage of paired vs unpaired males over the years. Thus, the y-axis does not show the percentage of paired males, but rather the percentage of males (where some were paired and some unpaired).
OUR RESPONSE: This has been changed. The x-axis now reads “Male age class (black head colour)” and the axis was changed form 0-5 to B0-B5; the y-axis is now “Pairing outcome (%).
Reviewer 2 Report
Dear Authors,
Thank you for providing a revised version of this manuscript. I The revised manuscript shows evidence that you have considered the feedback provided. The manuscript is more robust as a result.
Author Response
Thank you!